# The mass of the $\pi^+$

**M. Daum and P.-R. Kettle**

Paul Scherrer Institut, 5232 Villigen PSI, Switzerland

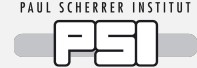

## Abstract

The most precise value for the pion mass was determined from a precision measurement at PSI of the muon momentum in pion decay at rest, $\pi^+ \rightarrow \mu^+ + \nu_\mu$. The result is $m_{\pi^+} = 139.570\,21(14)$ MeV/c². This value is more precise, however, in agreement with the recent compilation of the Particle Data Group for $m_{\pi^-}$. The agreement of $m_{\pi^+}$ with the recent measurement. This yields a new quantitative measure of CPT invariance in the pion sector: $(m_{\pi^+} - m_{\pi^-})/m_\pi(\text{av}) = (-2.9 \pm 2.0) \cdot 10^{-6}$, an improvement by two orders of magnitude.


## 11.1 Introduction

There has been a long-term effort at PSI to measure the momentum $p = |\vec{p}\,|$ of the muon from pion decay at rest [1–7],

$$\pi^+ \rightarrow \mu^+ + \nu_\mu. \tag{11.1}$$

Using energy and momentum conservation for the case of a pion at rest, its mass can be obtained as

$$m_{\pi^+} = \sqrt{m_{\mu^+}^2 + p^2} + \sqrt{m_{\nu_\mu}^2 + p^2}. \tag{11.2}$$

Assuming the validity of the CPT theorem, $m_{\pi^+} = m_{\pi^-}$, so this can also be written as

$$m_{\nu_\mu}^2 = m_{\pi^-}^2 + m_{\mu^+}^2 - 2m_{\pi^-}\sqrt{m_{\mu^+}^2 + p^2}. \tag{11.3}$$

The measurements of $p$ were originally intended to determine the mass of the muon neutrino, $m_{\nu_\mu}$, or its upper limit through (11.3). With stringent upper bounds on the neutrino mass from recent experiments of the neutrino sector, it is also possible to use (11.2) to obtain precise values for $m_{\pi^+}$ [8].

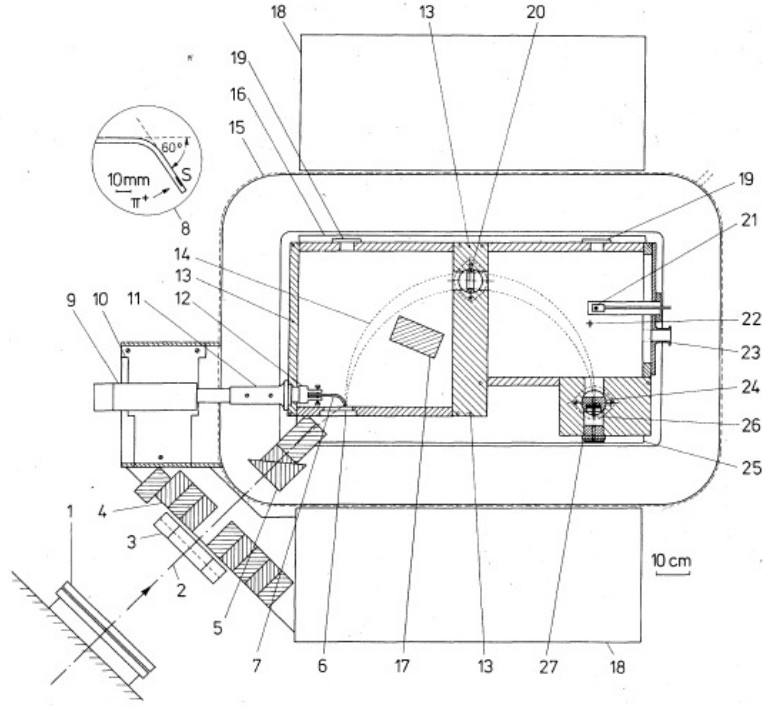

Figure 11.1: Experimental arrangement for the muon momentum measurement for Mark I-III: (1) exit vacuum window of the pion channel, (2) central trajectory of the pion beam, (3) multiwire proportional chamber for beam profile measurements, (4) lead collimator, (5) remotely controlled pion degrader, (6) window of the spectrometer vacuum chamber, (7) light guide of the pion-stop scintillation counter $S$, (8) enlarged view of the counter $S$, (9) photomultiplier of the counter $S$, (10) adjustable support of the photomultiplier, (11) vacuum feed through of the light guide, (12) positioning mechanism for the scintillator $S$, (13) vacuum chamber of the spectrometer, (14) region of accepted muon trajectories, (15) correction coils for magnetic field stabilization, (16) magnet pole, (17) beam stopper, (18) magnet yoke, (19) ports of the glass windows used for optical measurements of scintillator and collimator positions, (20) copper collimator, (21) NMR probe for magnetic field stabilization, (22) $^{241}$Am $\alpha$ source for the calibration of the silicon detector, (23) port for vacuum pump, (24) copper collimator, (25) magnet coils, (26) silicon surface barrier detector (Si) for muon detection, (27) coaxial vacuum feed-through for the counter Si.

## 11.2 Measurements at PSI

The measurement of the muon momentum in pion decay at rest was performed during five experimental periods (Mark I - V). A single focusing semicircular spectrometer with a homogeneous magnetic field was used. The experimental setup for Mark I-III is shown in Figure 11.1.

Positive pions of momentum 220 MeV/c enter the spectrometer and are slowed down in a degrader. A fraction of the pions stop in a small scintillator. The pions of interest are those that come to rest close to the downstream surface of the scintillator. Their decay muons can leave the scintillator with little or no energy loss. A muon created at the scintillator surface that starts along the central trajectory of the spectrometer, travels along this trajectory if the magnetic field is about 2760 Gauss. It is identified at the end of the trajectory by a silicon surface barrier detector. At higher magnetic fields the detected muon rate decreases to zero. At lower magnetic fields, detected muons come from a finite depth of the scintillator and

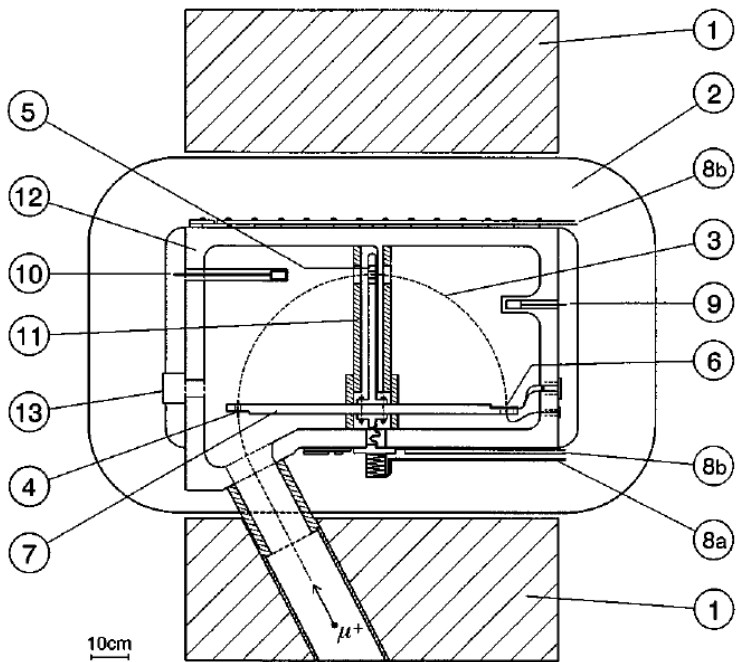

Figure 11.2: Experimental arrangement for the muon momentum measurement for Mark IV and V: (1) magnet yoke, (2) magnet coils, (3) central muon trajectory, (4)-(6) copper collimators, (7) titanium support (8a) and (8b) cooling water pipes, (9) and (10) NMR probes, (11) lead shielding, (12) vacuum chamber, (13) port for vacuum pumping.

therefore lose some of their energy before leaving the scintillator. Details of the apparatus and the analysis are described in [3].

The experimental setup for the Mark IV and V experiments is shown in Figure 11.2. In these experiments, a surface muon beam is used. The muons enter the spectrometer through a hole in the iron yoke of the spectrometer magnet. The angle between the axis of the hole and the outer surface of the yoke was 27°, chosen so that muons entering the hole on the axis have the appropriate flight direction at the entry collimator (item 4, Figure 11.2). These muons travel through the trajectory region and are detected in a position sensitive silicon microstrip detector behind a collimator (item 6, Figure 11.2). The 4.12 MeV muons lose about 0.9 MeV in passing through this detector and are then stopped in a 1 mm thick depletion layer of a single silicon surface barrier detector. The corresponding large signals from this latter detector were used as an event trigger for the data taking electronics. Details of the experiment and the analysis are described in [7].

The results from the five different experimental periods (Mark I to V) are given in Table 11.1. Initially, these results were used with (11.3) to obtain an upper limit on $m_{\nu_\mu}$. Using the known values for $m_{\mu^+}$ and $m_{\pi^-}$ at the time gives $m_{\nu_\mu}^2 = (-0.016 \pm 0.023) \, (\text{MeV}/c^2)^2$, which leads to an upper limit, $m_{\nu_\mu} \leq 170$ keV/$c^2$ with 90 % confidence [7]. Later, the accuracy of the $\pi^-$ and $\mu^+$ masses were improved [9–13]. These new mass values gives:

$$m_{\nu_\mu}^2 = (0.024 \pm 0.017) \, (\text{MeV}/c^2)^2 \tag{11.4}$$

which results in an upper limit with 90 % confidence,

$$m_{\nu_\mu} \leq 230 \text{ keV}/c^2. \tag{11.5}$$

Table 11.1: Results for the muon momentum from pion decay at rest. *This value includes the Mark I result [1–3]. **This value includes the Mark II result [4]. ***This value includes the Mark IV result [6].

| Mark | Year | $p$ [MeV/c] | Reference |
|---|---|---|---|
| I | 1979 | $29.7885 \pm 0.0019$ | [3,4] |
| II*) | 1984 | $29.79139 \pm 0.00083$ | [4] |
| III**) | 1991 | $29.79206 \pm 0.00068$ | [5] |
| IV | 1994 | $29.79207 \pm 0.00012$ | [6] |
| V***) | 1996 | $29.79200 \pm 0.00011$ | [7] |
| weighted mean | 2019 | $29.79200 \pm 0.00011$ | [8] |

An upper limit for the electron neutrino mass $m_{\nu_e}$ has been measured at the level of $m_{\nu_e} \leq 2$ eV/c$^2$ [11, 14, 15], and has recently been improved further [16]. This mass value represents the "effective" electron neutrino mass, which is the weighted sum of the mass eigenstates,

$$m_{\nu_e}^2 = \sum_{i=1}^{3} |U_{ei}|^2 m_{\nu_i}^2. \tag{11.6}$$

Here $U$ is the Pontecorvo–Maki–Nakagawa–Sakata matrix that relates the mass eigenstates $\nu_i$, $i = (1,2,3)$ to the flavor eigenstates $m_{\nu_e}$, $m_{\nu_\mu}$ and $m_{\nu_\tau}$. The mass differences $\Delta m_{21}$ and $\Delta m_{32}$ are experimentally found to be in the meV range [11, 17–20]. Consequently, the muon and tau neutrino masses must be equal to or less than $\sim 2$ eV/c$^2$. Thus, the measurements of the muon momentum from pion decay at rest can be re-interpreted as a precise direct determination of the mass of the positively charged pion, $m_{\pi^+}$.

According to (11.2), the uncertainty $\Delta m_{\pi^+}$ is limited by the uncertainties of $p$, $m_{\mu^+}$, and $m_{\nu_\mu}$. Taking the values $m_\mu = (105.6583745 \pm 0.0000024)$ MeV/c$^2$ [11–13] and (conservatively) $m_{\nu_\mu} = (2.0 \pm 2.0) \cdot 10^{-6}$ MeV/c$^2$, the total uncertainty is dominated by $p$. With the value as given in Table 11.1, the result for the mass of the positively charged pion is [8]

$$m_{\pi^+} = (139.57021 \pm 0.00014) \text{ MeV/c}^2. \tag{11.7}$$

While (11.7) is nearly the same value as published earlier [7], it is not affected by the limited knowledge of neutrino masses. In fact, the value of [7] was at the time interpreted as a lower limit on $m_{\pi^+}$, whereas now (11.7) is simply the most precise value for the charged pion mass with a precision of 1 ppm.

## 11.3 Summary of $m_\pi$ measurements at PSI

The measured values of $m_{\pi^-}$ from pionic atoms (see Section 10 [23]) and $m_{\pi^+}$ from our measurements are shown in Figure 11.3. The result (11.7) is more precise than and within 1.45 $\sigma$ of the recent compilation of the Particle Data Group (PDG) for $m_{\pi^\pm}$ [11]

$$m_{\pi^\pm} = (139.57061 \pm 0.00024) \text{ MeV/c}^2, \tag{11.8}$$

which uses the three most recent pionic atom experiments [9, 10, 22].[1] The agreement with the most precise single measurement of $m_{\pi^-}$ [10],

$$m_{\pi^-} = (139.57077 \pm 0.00018) \text{ MeV/c}^2 \tag{11.9}$$

---

[1] In fact, the Particle Data Group uses for their average only [9,10] and solution B of [22].

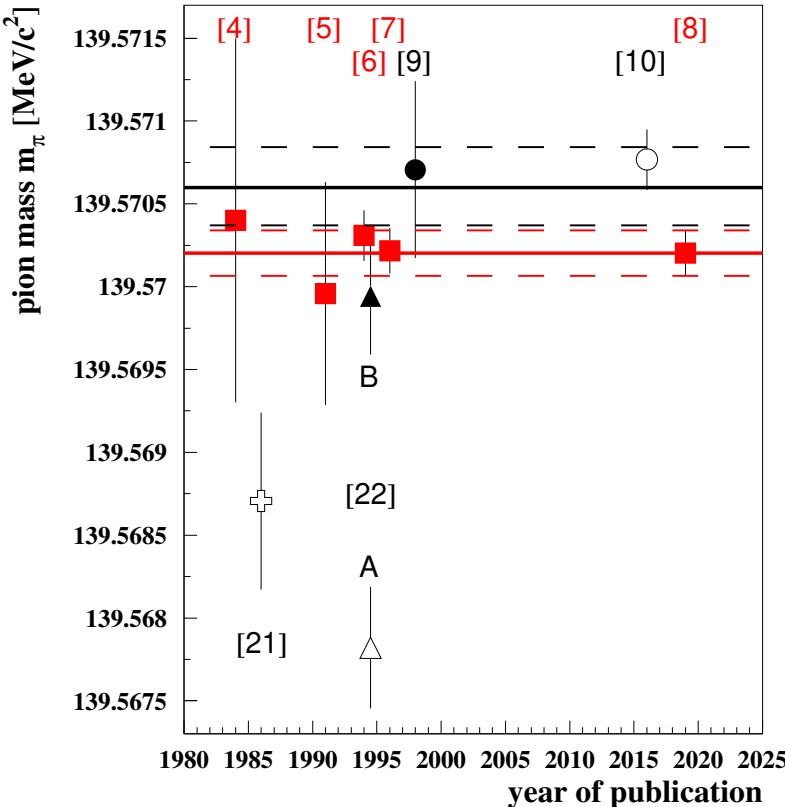

Figure 11.3: Plot of the evolution of the measured charged pion mass. Black symbols and lines: results for $m_{\pi^-}$ from pionic atoms. Red symbols and lines: results for $m_{\pi^+}$ from muon momentum in pion decay at rest. The $\pi^-$ measurements of [21] were re-analyzed after the $\pi^+$ results of [6] were published in view of the large discrepancy. The re-analysis resulted in two solutions in [22] A and B. The continuous and dashed black lines show the PDG average and 1 $\sigma$ band for the charged pion mass which comprises of purely pionic atom measurements: [22] solution B and [9,10], as earlier measurements and [22] solution A may have incorrect K-shell corrections [11]. The continuous and dashed red lines represent the final result, the weighted mean of our 1991 and 1996 values of $m_{\pi^+}$ together with the 1 $\sigma$ uncertainty band.

is only fair (2.4 $\sigma$)

$$m_{\pi^-} - m_{\pi^+} = (0.000\,56 \pm 0.000\,23) \text{ MeV/c}^2. \qquad (11.10)$$

Furthermore, by considering the masses of the positive and negative pion separately and comparing the PDG value, (11.8) which is based solely on $\pi^-$ measurements, with our $\pi^+$-value one has a quantitative measure of the CPT invariance in the pion sector. Using the PDG nomenclature one obtains

$$\frac{m_{\pi^+} - m_{\pi^-}}{m_{\text{av}}} = (-2.9 \pm 2.0) \cdot 10^{-6}. \qquad (11.11)$$

This is two orders of magnitude more precise than the best value so far, $(2 \pm 5) \cdot 10^{-4}$ [24]. Our result is consistent within 1.45 $\sigma$ with the CPT theorem.

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
