# Peer review of "The mass of the 4\pi^+$"

_SciPost Physics Proceedings, doi:SciPost Phys. Proc. 5, 011 (2021)_

## Round 1 · Referee Report · Anonymous (Referee 2) · 2021-7-5

Strengths

  1. detailed and well written review on the measurements at PSI of the \mu⁺ momentum after \pi⁺ decay at rest
  2. very informative figures 11.1 & 11.2 about the p_\mu⁺ experiments
  3. clear argumentation about neutrino mass upper limits and how to extract m_\pi⁺ from p_\mu⁺
  4. good figure 11.3 showing the time development of \pi⁺ / \pi⁻ mass results

Weaknesses

  1. the determinations of m_\pi⁻ are essentially only referenced and not much discussed.
  2. solution A and B only understandable for insiders. It is understood that PDG made the choice of B.

Report

The authors present a good review on the experiments resulting in high precision mass determinations of the charged pion masses m_\pi⁺ and m_\pi⁻. It is fascinating that by comparison of the two final values a measurement of CPT invariance at the 3 ppm level is obtained.
A discussion of the \pi⁻ mass measurements by pionic X ray experiments would be interesting, but is clearly beyond the scope of this paper.
This paper fits the criteria of the SciPost journal.

---

## Round 1 · Referee Report · Adrian Signer (Referee 1) · 2021-7-5

Report

We (the editors Cy Hoffman, Klaus Kirch, Adrian Signer) had the
opportunity to review an earlier draft of the article and were in
communication with the authors before the submission. All our comments
and suggestions have been taken into account. Hence, we think the
paper can now be published in the current form.

---

## Editorial Decision

published